# Adam with Bandit Sampling for Deep Learning

**Rui Liu, Tianyi Wu, Barzan Mozafari**
Computer Science and Engineering
University of Michigan, Ann Arbor
`{ruixliu, tianyiwu, mozafari}@umich.edu`

## Abstract

Adam is a widely used optimization method for training deep learning models. It computes individual adaptive learning rates for different parameters. In this paper, we propose a generalization of Adam, called ADAMBS, that allows us to also adapt to different training examples based on their importance in the model's convergence. To achieve this, we maintain a distribution over all examples, selecting a mini-batch in each iteration by sampling according to this distribution, which we update using a multi-armed bandit algorithm. This ensures that examples that are more beneficial to the model training are sampled with higher probabilities. We theoretically show that ADAMBS improves the convergence rate of Adam—$O(\sqrt{\frac{\log n}{T}})$ instead of $O(\sqrt{\frac{n}{T}})$ in some cases. Experiments on various models and datasets demonstrate ADAMBS's fast convergence in practice.

## 1 Introduction

Stochastic gradient descent (SGD) is a popular optimization method, which iteratively updates the model parameters by moving them in the direction of the negative gradient of the loss evaluated on a mini-batch. However, standard SGD does not have the ability to use past gradients or adapt to individual parameters (a.k.a. features). Some variants of SGD, such as AdaGrad [16], RMSprop [34], AdaDelta [36], or Nadam [15] can exploit past gradients or adapt to individual features. Adam [19] combines the advantages of these SGD variants: it uses momentum on past gradients, but also computes adaptive learning rates for each individual parameter by estimating the first and second moments of the gradients. This adaptive behavior is quite beneficial as different parameters might be of different importance in terms of the convergence of the model training. In fact, by adapting to different parameters, Adam has shown to outperform its competitors in various applications [19], and as such, has gained significant popularity.

However, another form of adaptivity that has proven beneficial in the context of basic SGD variants is adaptivity with respect to different examples in the training set [18, 6, 8, 38, 13]. For the first time, to the best of our knowledge, in this paper we show that accounting for the varying importance of training examples can even improve Adam's performance and convergence rate. In other words, Adam has only considered the varying importance among parameters but not among the training examples. Although some prior work exploits importance sampling to improve SGD's convergence rate [18, 6, 8, 38, 13], they often rely on some special properties of different models to estimate a sampling distribution over examples in advance. As a more general approach, we focus on learning the distribution during the training procedure, which is more adaptive. This is key to achieving faster convergence rate, because the sampling distribution typically changes from one iteration to the next, depending on the relationship between training examples and model parameters at each iteration.

In this paper, we propose a new general optimization method based on bandit sampling, called ADAMBS (Adam with Bandit Sampling), that endows Adam with the ability to adapt to different examples. To achieve this goal, we maintain a distribution over all examples, representing their

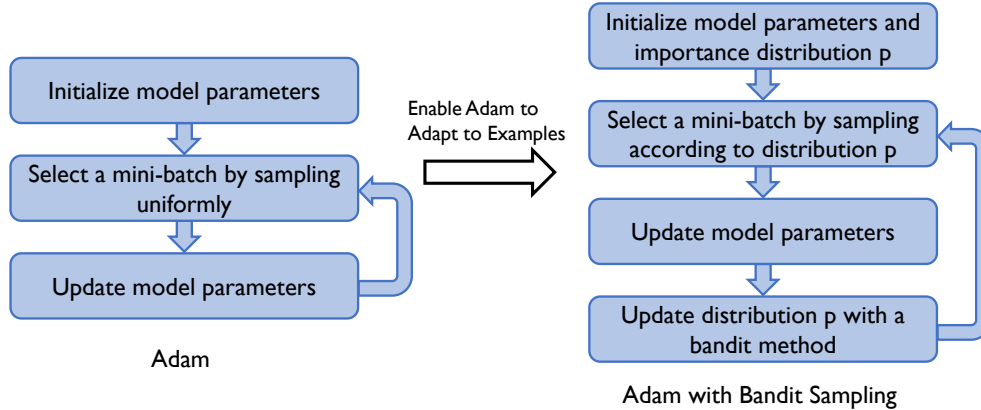

Figure 1: Adam with Bandit Sampling. Our method ADAMBS generalizes Adam with the ability to adapt to different examples by using a bandit method to maintain a distribution $p$ over all training examples.

relative importance to the overall convergence. In Adam, at each iteration, a mini-batch is selected *uniformly* at random from training examples. In contrast, we select our mini-batch according to our maintained distribution. We then use this mini-batch to compute the first and second moments of the gradient and update the model parameters, in the same way as Adam. While seemingly simple, this process introduces another challenge: how to efficiently obtain this distribution and update it at each iteration? Ideally, to obtain the optimal distribution, at each iteration one would need to compute the gradients for all training examples. However, in Adam, we only have access to gradients of the examples in the selected mini-batch. Since we only have access to partial feedback, we use a multi-armed bandit method to address this challenge. Our idea is illustrated in Figure 1.

Specifically, we cast the process of learning an optimal distribution as an adversarial multi-armed bandit problem. We use a multi-armed bandit method to update the distribution over all of the training examples, based on the partial information gained from the mini-batch at each iteration. We use the EXP3 method [3], but extend it to allow for sampling multiple actions at each step. The original EXP3 method only samples one action at a time, and collects the partial feedback by observing the loss incurred by that single action. In contrast, in optimization frameworks such as Adam, we need to sample a mini-batch, which typically contains more than one example. We thus need a bandit method that samples multiple actions and observes the loss incurred by each of those actions. In this paper, we extend EXP3 to use feedback from multiple actions and update its distribution accordingly. Although ideas similar to bandit sampling have been applied to some SGD variants and coordinate descent methods [30, 29, 24], extending this idea to Adam is not straightforward, as Adam is considerably more complex, due to its momentum mechanism and parameter adaptivity. To the best of our knowledge, we are the first to propose and analyze the improvement of using bandit sampling for Adam. Maintaining and updating distribution over all training examples incur some computational overhead. We will show an efficient way to update this distribution, which has time complexity logarithmic with respect to the total number of training examples. With this efficient update, the per-iteration cost is dominated by gradient computation, whose time complexity depends on the mini-batch size.

To endow Adam with the adaptive ability to different examples while keeping its original structure, we interleave our bandit method with Adam's original parameter update, except that we select the mini-batch according to our maintained distribution. ADAMBS therefore adapts to both different parameters and different examples. We provide a theoretical analysis of this new method showing that our bandit sampling does indeed improve Adam's convergence rate. Through an extensive empirical study across various optimization tasks and datasets, we also show that this new method yields significant speedups in practice as well.

## 2   Related Work

**Boosting and bandit methods**.   The idea of taking advantage of the difference among training examples has been utilized in many boosting algorithms [32, 33, 31, 22, 12].  The well-known

AdaBoost algorithm [32] builds an ensemble of base classifiers iteratively, of which each base classifier is trained on the same set of training examples with adjusted weights. Because of the way AdaBoost adjusts weights on training examples, it is able to focus on examples that are hard, thus decreasing the training error very quickly. In addition, it has been often observed in experiments that AdaBoost has very good generalization ability, which is discussed and analyzed in several work [33, 28, 22]. Both AdaBoost and our method aim to improve training by iteratively adjusting example weights. However, the amount of available information is very different every time they adjust example weights. AdaBoost receives full information, in the sense that each training example needs to run through the up-to-date ensemble model to determine which examples are still misclassified. Our method only receives partial information because we can only select a mini-batch at each iteration, which brings up the tradeoff between exploration (i.e., select other examples to get more information) and exploitation (i.e., select the empirically best examples based on already collected information). Multi-armed bandit problem is a general setting for studying the exploration-exploitation tradeoff that also appears in many other cases [4, 3]. For example, it has been applied to speed up maximum inner product search when only a subset of vector coordinates can be selected for floating point multiplication at each round [23].

**Importance sampling methods**. Importance sampling for convex optimization problems has been extensively studied over the last few years. [27] proposed a generalized coordinate descent algorithm that samples coordinate sets to optimize the algorithm's convergence rate. More recent works [38, 25] discuss the variance of the gradient estimates of stochastic gradient descent and show that the optimal sampling distribution is proportional to the per-sample gradient norm. [24] proposed an adaptive sampling method for both block coordinate descent and stochastic gradient descent. For coordinate descent, the parameters are partitioned into several prefixed blocks; for stochastic gradient descent, the training examples are partitioned into several prefixed batches. However, it is difficult to determine an effective way to partition blocks of parameters or batches of examples. [18] proposed to sample a big batch at every iteration to compute a distribution based on gradient norms of these examples from the big batch, followed by a mini-batch that is sampled from the big batch for parameter update. However, it is unclear how much speedup their method can achieve in terms of theoretical convergence rate.

**Other sample selection methods**. Several strategies have been proposed to carefully select mini-batches in order to improve on training deep learning models. Curriculum learning [5, 17] is another optimization strategy that leverages a pre-trained teacher model to train the target model. The teacher model is critical in determining how to assign examples to mini-batches. In this paper, we focus on the case when we do not have access to an extra teacher model. However, utilizing a teacher model is likely to further improve the performance of our method. For example, it can be used to initialize the example weights which can help the bandit method to learn the weights more quickly. In addition, Locality-Sensitive Hashing (LSH) has been used to improve the convergence rate of SGD by adaptively selecting examples [9]. It is worth noting that a recent paper [26] points out that Adam's rapid decay of the learning rate using the exponential moving averages of squared past gradients essentially limits the reliance of the update to only the past few gradients. This prevents Adam from convergence in some cases. They proposed a variant of Adam, called AMSGRAD, with long term memory of past gradients. In the main text of this paper, we remain focused on the original Adam. Similar analysis could be carried over to AMSGRAD, which is discussed in appendix. We note that Adam with the learning rate warmup (another variant of Adam), is the common practice in training transformer models for NLP tasks [14, 11]. However, due to the lack of well-studied theoretical analysis of this variant in the literature, we still base our dicussion on the original Adam.

## 3 Preliminaries about Adam

We consider the following convex optimization problem: $\min_{\theta \in \mathcal{X}} f(\theta)$ where $f$ is a scalar-valued objective function that needs to be minimized, and $\theta \in \mathbb{R}^d$ is the parameter of the model. Let the gradient of $f$ with respect to $\theta$ be denoted as $G$. Assuming the training dataset is of size $n$, we have $G = \frac{1}{n} \sum_{i=1}^{n} g_i$, where $g_i$ is the gradient computed with only the $i$-th example. Furthermore, at each iteration $t$, we select a mini-batch of examples from the whole training set. We denote the realization of $f$ with respect to the mini-batch selected at iteration $t$ as $f_t$, and the gradient of $f_t$ with respect to $\theta$ as $G_t$. Depending on the sampling strategy of a mini-batch, in some cases, $G_t$ could be a biased estimate of $G$, meaning $\mathbb{E}[G_t] \neq G$. However, an unbiased estimate is required to update model parameters in stochastic optimization such as SGD and Adam. In such cases, we need to

get an unbiased estimate $\hat{G}_t$, ensuring that $\mathbb{E}[\hat{G}_t] = G$. When a mini-batch is selected by sampling uniformly from all of the training examples, we have $\mathbb{E}[G_t] = G$, thus allowing $\hat{G}_t$ to simply be $G_t$.

Adam [19] selects every mini-batch by uniform sampling and updates exponential moving averages of the gradient $m_t$ and the squared gradient $v_t$ with hyperparameters $\beta_1, \beta_2 \in [0, 1)$, which control the exponential decay rates of these moving averages: $m_t \leftarrow \beta_1 \cdot m_{t-1} + (1 - \beta_1) \cdot \hat{G}_t$, $v_t \leftarrow \beta_2 \cdot v_{t-1} + (1 - \beta_2) \cdot \hat{G}_t^2$ where $\hat{G}_t^2$ indicates the element-wise square of $\hat{G}_t$. The moving averages $m_t$ and $v_t$ are estimates of the 1st moment (the mean) and 2nd raw moment (the uncentered variance) of the gradient. These moment estimates are biased toward zero and are then corrected, resulting in bias-corrected estimates $\hat{m}_t$ and $\hat{v}_t$: $\hat{m}_t \leftarrow m_t/(1 - \beta_1^t)$, $\hat{v}_t \leftarrow v_t/(1 - \beta_2^t)$ where $\beta_1^t$ and $\beta_2^t$ are $\beta_1$ and $\beta_2$ raised to the power $t$, respectively. Next, the parameter is updated according to $\theta_t \leftarrow \theta_{t-1} - \alpha_\theta \cdot \hat{m}_t/(\sqrt{\hat{v}_t} + \epsilon)$ where $\epsilon$ is a small value, to avoid division by zero.

A flexible framework to analyze iterative optimization methods such as Adam is the online learning framework. In this online setup, at each iteration $t$, the optimization algorithm picks a point $\theta_t \in \mathbb{R}^d$. A loss function $f_t$ is then revealed based on the seleted mini-batch, and the algorithm incurs loss $f_t(\theta_t)$. At the end of $T$ iterations, the algorithm's regret is given by $R(T) = \sum_{t=1}^{T} f_t(\theta_t) - \min_{\theta \in \mathcal{X}} \sum_{t=1}^{T} f_t(\theta)$. In order for any optimization method to converge, it is necessary to ensure that $R(T) = o(T)$. For Adam, the convergence rate is summarized in the Theorem 4.1 from [19]. Under further assumptions as in Corollary 4.2 from [19], it can be shown that $R(T) = o(T)$. In this paper, we propose to endow Adam with bandit samping which could further improve the convergence rate under some assumptions.

# 4 Adam with Bandit Sampling

## 4.1 Adaptive Mini-Batch Selection

Suppose there are $n$ training examples. At iteration $t$, a mini-batch of size $K$ is selected by sampling with replacement from all of the training examples, according to a distribution $p^t = \{p_1^t, p_2^t, \cdots, p_n^t\}$. Here, $p^t$ represents the relative importance of each example during the model training procedure. Denote the indices of examples selected in the mini-batch as the set $I^t = \{I_1^t, I_2^t, \cdots, I_K^t\}$. Assume the gradient computed with respect to the only example $I_k^t$ is $g_{I_k^t}$. Its unbiased estimate is $\hat{g}_{I_k^t} = \frac{g_{I_k^t}}{n p_{I_k^t}}$.

We can easily verify that $\hat{g}_{I_k^t}$ is unbiased because $\mathbb{E}_{p^t}\left[\hat{g}_{I_k^t}\right] = \frac{1}{n} \sum_{i=1}^{n} g_i = G$. Therefore, we define the unbiased gradient estimate $\hat{G}_t$ according to batch $I^t$ at iteration $t$ as

$$\hat{G}_t = \frac{1}{K} \sum_{k=1}^{K} \hat{g}_{I_k^t}. \tag{1}$$

Similarly, we can verify that $\mathbb{E}_{p^t}\left[\hat{G}_t\right] = \frac{1}{n} \sum_{i=1}^{n} g_i = G$.

It is worth noting that $\hat{G}_t$ defined for ADAMBS is different than that of Adam. This is because the sampling strategy is different, and appropriate bias-correction is necessary here. We use $\hat{G}_t$, defined in Equation 1, to update first moment estimate and second moment estimate in each iteration. Specifically, let $m_t$ and $v_t$ be the first and second moment estimates at iteration $t$, respectively. Then we update them in the following way

$$m_t = \beta_1 \cdot m_{t-1} + (1 - \beta_1) \cdot \hat{G}_t, \ \ v_t = \beta_2 \cdot v_{t-1} + (1 - \beta_2) \cdot \hat{G}_t^2 \tag{2}$$

where $\beta_1, \beta_2 \in [0, 1)$ are hyperparameters that control the exponential decay rates of these moving averages. Our new method ADAMBS is described in Algorithm 1. Details about *update* function in line 11 are given in the next subsection.

Our method mantains a fine-grained probability distribution over all examples. This provides more flexibility in choosing mini-batches than prior work that uses coarse-grained probability distribution over pre-fixed mini-batches [24], because it is generally hard to decide how to partition the batches for pre-fixed mini-batches. If the training set is partitioned randomly, any mini-batch is likely to contain some important examples and some unimportant examples, making any two mini-batches equally good. In this case, prioritizing one mini-batch over another will not bring any advantage. It requires

**Algorithm 1** ADAMBS, our proposed Adam with bandit sampling. Note that $\beta_1, \beta_2$ are hyperparameters controling exponential decay rates of first and second moment estimates, respectively. $\beta_1^t, \beta_2^t$ denote $\beta_1, \beta_2$ raised to power $t$, respectively. $\alpha_\theta$ is the learning rate for model parameter update. $\hat{G}_t^2$ is the element-wise square of $\hat{G}_t$.

---

1: Initialize model parameter $\theta_0$;
2: Initialize first moment estimate $m_0 \leftarrow 0$, and second moment estimate $v_0 \leftarrow 0$;
3: Initialize distribution $p_j^0 \leftarrow \frac{1}{n}, \forall 1 \leq j \leq n$, and set iteration index $t = 0$;
4: **while** $\theta_t$ not converged **do**
5:    $t \leftarrow t + 1$;
6:    select a mini-batch $I^t$ by sampling with replacement from $p^{t-1}$;
7:    compute unbiased gradient estimate $\hat{G}_t$;
8:    $m_t \leftarrow \beta_1 \cdot m_{t-1} + (1 - \beta_1) \cdot \hat{G}_t, \quad v_t \leftarrow \beta_2 \cdot v_{t-1} + (1 - \beta_2) \cdot \hat{G}_t^2$;
9:    $\hat{m}_t \leftarrow m_t/(1 - \beta_1^t), \quad \hat{v}_t \leftarrow v_t/(1 - \beta_2^t)$;
10:   $\theta_t \leftarrow \theta_{t-1} - \alpha_\theta \cdot \hat{m}_t/(\sqrt{\hat{v}_t} + \epsilon)$;
11:   $p^t \leftarrow update(p^{t-1}, I^t, \{g_{I_k^t}\}_{k=1}^K)$;
12: **end while**

---

a fair amount of time on preprocessing the training dataset to partition the dataset in a good way, especially when the dataset is large. Some might argue that we could simply set the batch size to one in [24]. While the issue of batch partitioning does not exist anymore, this would significantly hurt the convergence rate because only one example is processed at each iteration. In contrast, ADAMBS does not require pre-partitioning mini-batches. At every iteration, a new mini-batch is formed dynamically by sampling from the whole dataset according to distribution $p^{t-1}$. Here, the distribution $p^{t-1}$ is learned so that important examples can be selected into one mini-batch with high probability. Thus, it is more likely to get a mini-batch with all important examples, which could significantly boost the training performance.

We analyze the convergence of ADAMBS in Algorithm 1 using the same online learning framework [39] that is used by Adam. The following theorem [1] holds, regarding the convergence rate of ADAMBS.

**Theorem 1.** *Assume that the gradient estimate $\hat{G}_t$ is bounded, $\|\hat{G}_t\|_2 \leq G, \|\hat{G}_t\|_\infty \leq G_\infty$, and distance between any $\theta_t$ is bounded, $\|\theta_n - \theta_m\|_2 \leq D, \|\theta_n - \theta_m\|_\infty \leq D_\infty$ for any $m, n \in \{1, \cdots, T\}$, and $\beta_1, \beta_2 \in [0, 1)$ satisfy $\frac{\beta_1^2}{\sqrt{\beta_2}} < 1$. Let $\alpha_\theta = \frac{\alpha}{\sqrt{t}}$, and $\beta_{1,t} = \beta_1 \lambda^{t-1}, \lambda \in (0, 1)$, and $\gamma = \frac{\beta_1^2}{\sqrt{\beta_2}}$. ADAMBS achieves the following convergence rate, for all $T \geq 1$,*

$$R(T) \leq \rho_1 d\sqrt{T} + \sqrt{d}\rho_2 \sqrt{\frac{1}{n^2 K} \sum_{t=1}^T \mathbb{E}\left[\sum_{j=1}^n \frac{\|g_j^t\|^2}{p_j^t}\right]} + \rho_3 \tag{3}$$

*where $d$ is the dimension of parameter space and*

$$\rho_1 = \frac{D^2 G_\infty}{2\alpha(1 - \beta_1)}, \quad \rho_2 = \frac{\alpha(1 + \beta_1)G_\infty}{(1 - \beta_1)\sqrt{1 - \beta_2}(1 - \gamma)^2}, \quad \rho_3 = \sum_{i=1}^d \frac{D_\infty^2 G_\infty \sqrt{1 - \beta_2}}{2\alpha(1 - \beta_1)(1 - \gamma)^2}. \tag{4}$$

### 4.2 Update of Distribution $p^t$

From Equation 3, we can see that $p^t$ will affect the convergence rate. We wish to choose $p^t$ that could lead to a faster convergence rate. We derive how to update $p^t$ by minimizing the right side of Equation 3. Specifically, we want to minimize $\sum_{t=1}^T \mathbb{E}\left[\sum_{j=1}^n \frac{\|g_j^t\|^2}{p_j^t}\right]$. It can be shown that for every iteration $t$, the optimal distribution $p^t$ is proportional to the gradient norm of the individual example [25, 2]. Formally speaking, for any $t$, the optimal solution to the problem $\arg\min_{\sum_{i=1}^n p_i^t = 1} \sum_{j=1}^n \frac{\|g_j^t\|^2}{p_j}$ is $p_j^t = \frac{\|g_j^t\|}{\sum_{i=1}^n \|g_i^t\|}, \forall j$. It is computationally prohibitive to get the optimal solution, because we need to compute the gradient norm for every example at each iteration. Instead, we use a multi-armed

bandit method to learn this distribution from the partial information that is available during training. The multi-armed bandit method maintains the distribution over all examples, and keeps updating this distribution at every training iteration. The partial information that we have at every iteration is the gradients of examples in the mini-batch. We use a bandit method based on EXP3 [3] but extended to handle multiple actions at every iteration. The pseudocode is described in Algorithm 2.

---

**Algorithm 2** The distribution update rule for $p^t$.

---

1: **Function**: $update(p^{t-1}, I^t, \{g_{I_k^t}\}_{k=1}^K)$
2:     Compute loss $l_{t,j} = -\frac{\|g_j^t\|^2}{(p_j^t)^2} + \frac{L^2}{p_{min}^2}$ if $j \in I^t$; otherwise, $l_{t,j} = 0$;
3:     Compute an unbiased gradient estimate $\hat{h}_{t,j} = \frac{l_{t,j} \sum_{k=1}^K \mathbb{1}(j=I_k^t)}{Kp_j^t}, \forall 1 \le j \le n$;
4:     $w_j^t = p_j^{t-1} \exp(-\alpha_p \hat{h}_{t,j}), \forall 1 \le j \le n$;
5:     $p^t = \arg\min_{q \in \mathcal{P}} D_{kl}(q\|w^t)$;
6: **Return:** $p^t$

---

To further illustrate our distribution update algorithm from the perspective of the bandit setting, the number of arms is $n$, where each arm corresponds to each training example, and the loss of pulling the arm $j$ is $l_{t,j}$ which is defined at line 2. We denote $L$ as the upper bound on the per-example gradient norm, i.e., $\|g_j^t\| \le L, \forall t, j$. Similar upper bound is commonly used in related literature [19, 24], because of the popular gradient clipping trick [37]. Each time we update the distribution, we only pull these arms from the set $I^t$, which is the mini-batch of training examples at current iteration. In line 5, $D_{kl}(q\|w^t)$ is the KL divergence between $q$ and

$w^t$, and the set $\mathcal{P}$ is defined as $\{p \in \mathbb{R}^n : \sum_{j=1}^n p_j = 1, p_j \ge p_{min} \forall j\}$, where $p_{min}$ is a constant. From the definition of $l_{t,j}$ as in line 2 from Algorithm 2, we can see that the loss $l_{t,j}$ is always nonnegative, and is inversely correlated with the gradient norm $\|g_j^t\|$. This implies that an example with small gradient norm will receive large loss value, resulting in its weight getting decreased. Thus, examples with large gradient norms will be sampled with higher probabilities in subsequent iterations. Using a segment tree structure to store the weights for all the examples, we are able to update $p^{t-1}$ in $O(K \log n)$ time [24]. With the efficient way to update distribution, the per-iteration cost is dominated by the computation of gradients, especially for large models in deep learning. For simplicity, our theoretical analysis is still focused on convergence rate with respect to the number of iterations. In experiments, we demonstrate that our method can achieve a faster convergence rate with respect to time.

Let $B_\phi(x, y) = \phi(x) - \phi(y) - \langle \nabla\phi(y), x - y \rangle$ be the Bregman divergence associated with $\phi$, where $\phi(p) = \sum_{j=1}^n p_j \log p_j$. By choosing this form of Bregman divergence, we can now study our EXP3-based distribution update algorithm under the general framework of online mirror descent with bandit feedback [7]. We have the following lemma regarding the convergence rate of Algorithm 2.

**Lemma 1.** *Assume that $\|g_j^t\| \le L, p_j^t \ge p_{min}$ for all $t$ and $j$, and $B_\phi(p, p') \le R^2$ for any $p, p' \in \mathcal{P}$, if we set $\alpha_p = \sqrt{\frac{2R^2 p_{min}^4}{nTL^4}}$, the update in Algorithm 2 implies*

$$\mathbb{E}\sum_{t=1}^T \sum_{j=1}^n \frac{\|g_j^t\|^2}{p_j^t} - \min_{p \in \mathcal{P}} \mathbb{E}\sum_{t=1}^T \sum_{j=1}^n \frac{\|g_j^t\|^2}{p_j} \le \frac{RL^2}{p_{min}^2}\sqrt{2nT}. \tag{5}$$

Combining Theorem 1 and Lemma 1, we have the following theorem regarding the convergence of ADAMBS.

**Theorem 2.** *Under assumptions from both Theorem 1 and Lemma 1,* ADAMBS *with the distribution update rule in Algorithm 2 achieves the following convergence rate*

$$R(T) \le \rho_1 d\sqrt{T} + \rho_2 \frac{\sqrt{d}}{n\sqrt{K}}\sqrt{M} + \frac{\rho_2 L\sqrt{R}}{p_{min}}\frac{\sqrt{d}}{n\sqrt{K}}(2nT)^{1/4} + \rho_3 \tag{6}$$

*where $M = \min_p \sum_{t=1}^T \mathbb{E}\left[\sum_{j=1}^n \frac{\|g_j^t\|^2}{p_j}\right]$, and $\rho_1, \rho_2, \rho_3$ are defined in Equation 4.*

Obviously, we have that, for ADAMBS, $R(T) = o(T)$, ensuring that it can converge.

### 4.3 Comparison with Uniform Sampling

We now study a setting where we can further bound Equation 6 to show that the convergence rate of our ADAMBS is provably better than that of Adam, which uses uniform sampling. For

Table 1: Convergence rates under doubly heavy-tailed distribution.

| Algorithm | Convergence Rate | Algorithm | Convergence Rate |
|---|---|---|---|
| ADAMBS | $O(d\sqrt{T}) + O(\frac{\sqrt{d\log d}}{\sqrt{K}}\frac{\sqrt{\log^2 n}}{n}\sqrt{T})$ | ADAM-APT | $O(d\sqrt{T}) + O(\sqrt{d\log d}\frac{\sqrt{\log^2 n}}{n}\sqrt{T})$ |
| Adam | $O(d\sqrt{T}) + O(\frac{\sqrt{d\log d}}{\sqrt{K}}\frac{\sqrt{n\log n}}{n}\sqrt{T})$ | ADAM-UNI | $O(d\sqrt{T}) + O(\sqrt{d\log d}\frac{\sqrt{n\log n}}{n}\sqrt{T})$ |

simplicity, we consider a neural network with one hidden layer for a binary classification problem. The hidden layer contains one neuron with ReLU activation, and output layer also contains one neuron with sigmoid activation. Cross-entropy loss is used for this binary classification problem. The total loss of this neural network model can be written in the following way $f(\theta) = \frac{1}{n}\sum_{j=1}^{n} f_j(\theta) = -\frac{1}{n}\sum_{j=1}^{n} y_j \log \sigma(ReLU(z_j^T\theta))$ where $\sigma(\cdot)$ is the sigmoid activation function, $ReLU(\cdot)$ is the ReLU activation function, $y_j \in \{-1, +1\}$ and $z_j$ are the label and the feature vector for $j$-th example, respectively. Denote that $\theta^* = \arg\min f(\theta)$. Here, $g_j = \partial f_j(\theta) = -y_j(1-\sigma(ReLU(z_j^T\theta)))\mathbb{1}(z_j^T\theta > 0)z_j$. It implies that $\|g_j\|^2 \leq \|z_j\|^2 = \sum_{i=1}^{d} z_{j,i}^2$. We further asssume that feature vector follows doubly heavy-tailed distribution, which means that, $z_{j,i} \in \{-1, 0, +1\}$ and $\mathbb{P}(|z_{j,i}| = 1) = \beta_3 i^{-\gamma} j^{-\gamma}$, where $\gamma \geq 2$.

**Lemma 2.** *If the examples are sampled with uniform distribution, i.e. $p_j = 1/n, \forall 1 \leq j \leq n$, assuming that feature vector follows doubly heavy-tailed distribution, for the aforementioned neural network model, we have $\mathbb{E}\sum_{j=1}^{n}\frac{\|g_j\|^2}{p_j} = \beta_3 n \log n \log d$.*

Following Lemma 2, we have

**Theorem 3.** *Assuming that feature vector follows doubly heavy-tailed distribution, for the aforementioned neural network model, original Adam achieves the following rate*

$$R(T) \leq O(d\sqrt{T}) + O(\frac{\sqrt{d\log d}}{\sqrt{K}}\frac{\sqrt{n\log n}}{n}\sqrt{T}). \tag{7}$$

On the other hand, we have

**Lemma 3.** *Assuming that feature vector follows doubly heavy-tailed distribution, for the aforementioned neural network model, we have*

$$\min_{p_j \geq p_{min}} \sum_{j=1}^{n}\frac{\|g_j\|^2}{p_j} = O(\log d \log^2 n). \tag{8}$$

By plugging Lemma 3 into Theorem 2, we have

**Theorem 4.** *Assuming that feature vector follows doubly heavy-tailed distribution, for the aforementioned neural network model, ADAMBS achieves the following rate*

$$R(T) \leq O(d\sqrt{T}) + O(\frac{\sqrt{d\log d}}{\sqrt{K}}\frac{\sqrt{\log^2 n}}{n}\sqrt{T}). \tag{9}$$

Comparing the second terms at Equations 7 and 9, we see that our ADAMBS converges faster than Adam. The convergence rates are summarized in Table 1. In this table, we also compare against adaptive sampling methods from [24]. They maintain a distribution over prefixed batches. Adam with adaptive sampling over prefixed batches is called ADAM-APT, and Adam with unifom sampling over prefixed batches is called ADAM-UNI. We can see that our method ADAMBS achieves faster convergence rate than the others. Depending on constants not shown in the big-$O$ notation, it's also possible that the convergence rate is dominated by the first term $O(d\sqrt{T})$, which makes our improvement marginal. We also rely on the experiments in the next section to demonstrate our method's faster convergence rate in practice.

## 5 Experiments

### 5.1 Setup

To empirically evaluate the proposed method, we investigate different popular deep learning models. We use the same parameter initialization when comparing different optimization methods. In total, 5

datasets are used: MNIST, Fashion MNIST, CIFAR10, CIFAR100 and IMDB. We run experiments on these datasets because they are benchmark datasets commonly used to compare optimization methods for deep learning [19, 26, 18]. It is worth noting that the importance sampling method proposed in [18] could also be applied to Adam. In addition, they proposed an efficient way to upper bound the per-example gradient norm for neural networks to compute the distribution for importance sampling. This could also be beneficial to our method, because the upper bound of gradient norm could be used in place of the gradient norm itself to update our distribution. In the experiments, we compare our method against Adam and Adam with importance sampling (as described in [18], which we call ADAM-IMPT). To be fair, we use the upper bound in the place of per-example gradient norm in our method. All the previous analysis also holds if an upper bound of gradient norm is used, because similar to Theorem 1, it will still upper bound $R(T)$. Experiments are conducted using Keras [10] with TensorFlow [1] based on the code from [18]. To see if our method could accelerate the training procedure, we plot the curves of training loss value vs. wall clock time for these three methods [2]. The curves are based on average results obtained by repeating each experiment five times with different random seeds. To see the difference in the performance of these optimization method, we use the same values for hyperparameters. Specifically, $\beta_1$ and $\beta_2$ are common hyperparameters to Adam, ADAM-IMPT and ADAMBS, which are chosen to be $0.9$ and $0.999$, respectively.

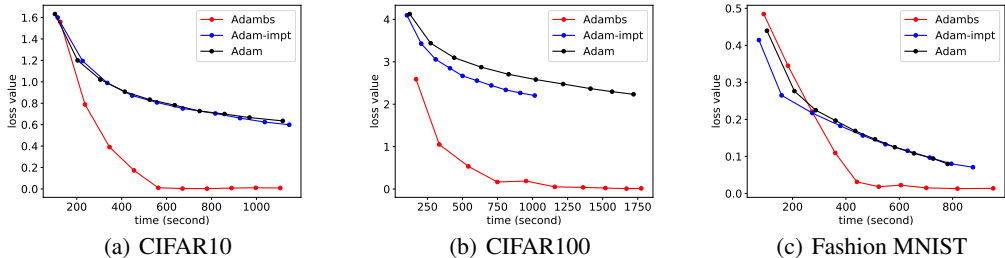

| (a) CIFAR10 | (b) CIFAR100 | (c) Fashion MNIST |

Figure 2: CNN model on CIFAR10, CIFAR100 and Fashion MNIST

## 5.2 Convolutional Neural Networks

Convolutional neural networks (CNN) with several layers of convolution, pooling, and non-linear units have shown considerable success in computer vision tasks. We train CNN models on three different datasets: CIFAR10, CIFAR100 and Fashion MNIST. CIFAR10 and CIFAR100 are labeled subsets of the 80 million tiny images dataset [20]. CIFAR10 consists of $60,000$ color images of size $32 \times 32$ in 10 classes with $6,000$ images per class, whereas CIFAR100 consists of $60,000$ color images of size $32 \times 32$ in 100 classes with $600$ images per class. Fashion MNIST dataset [35] is similar to MNIST dataset except that images are in $10$ fashion categories.

For CIFAR10 and CIFAR100, our CNN architecture has $4$ layers of $3 \times 3$ convolution filters and max pooling with size $2 \times 2$. Dropout with dropping probability $0.25$, is applied to the 2nd and 4th convolutional layers. This is then followed by a fully connected layer of $512$ hidden units. For Fashion MNIST, since the dataset is simpler, we use a simpler CNN model. It contains 2 layers of $3 \times 3$ filters and max pooling with size $2 \times 2$ is applied to the 2nd convolutional layer, which is followed by a fully connected layer of $128$ hidden units. The mini-batch size is set to $128$, and learning rate is set to $0.001$ for all methods on all three datasets. All three methods are used to train CNN models for 10 epochs and the results are shown in Figure 2. For CIFAR10 and CIFAR100, we can see that our method ADAMBS achieved loss value lower than others very quickly, within $1$ or $2$ epochs. For Fashion MNIST, our method ADAMBS is worse than others at the very begining, but keeps decreasing the loss value at a faster rate. After around $300$ seconds, ADAMBS is able to achieve lower loss value than others.

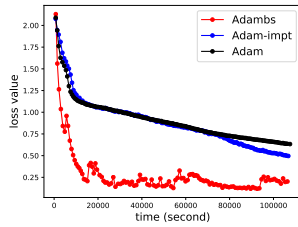
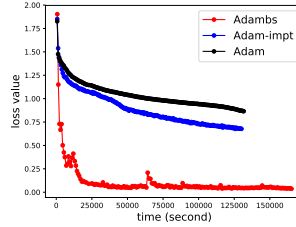
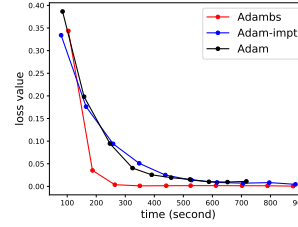

(a) MNIST　　　　　　(b) Fashion MNIST

Figure 3: RNN model on MNIST and Fashion MNIST

Figure 4: RCNN model on IMDB

## 5.3　Recurrent Neural Networks

Recurrent neural networks, such as LSTM and GRU, are popular models to learn from sequential data. To showcase the generality of our method, we use ADAMBS to accelerate the training of RNN models in image classification problems, where image pixels are fed to RNN as a sequence. Specifically, we train an LSTM with dimension $100$ in the hidden space, and ReLU as recurrent activation function, which is followed by a fully connected layer to predict image class. We use two datasets: MNIST and Fashion MNIST. The batch size is set as $32$, the learning rate is set as $10^{-6}$, and the maximum number of epochs is set as $200$ for all methods on both datasets. The results are shown in Figure 3. ADAMBS was able to quickly achieve lower loss value than the others.

## 5.4　Recurrent Convolutional Neural Networks

Recently, it has been shown that RNN combined with CNN can achieve good performance on some NLP tasks [21]. This new architecture is called recurrent convolutional neural network (RCNN). We train an RCNN model for the sentiment classification task on an IMDB movie review dataset. It contains $25,000$ movie reviews from IMDB, labeled by sentiment (positive or negative). Reviews are encoded by a sequence of word indexes. On this dataset, we train an RCNN model, which consists of a convolutional layer with filter of size $5$, and a max pooling layer of size $4$, followed by an LSTM and a fully connected layer. We set batch size to $30$ and learning rate to $0.001$, and run all methods for $10$ epochs. The result is shown in Figure 4. We can see that all methods converge to the same loss value, but ADAMBS arrives at convergence much faster than the others.

## 6　Conclusion

We have presented an efficient method for accelerating the training of Adam by endowing it with bandit sampling. Our new method, ADAMBS, is able to adapt to different examples in the training set, complementing Adam's adaptive ability for different parameters. A distribution is maintained over all examples and represents their relative importance. Learning this distribution could be viewed as an adversarial bandit problem, because only partial information is available at each iteration. We use a multi-armed bandit approach to learn this distribution, which is interleaved with the original parameter update by Adam. We provided a theoretical analysis to show that our method can improve the convergence rate of Adam in some settings. Our experiments further demonstrate that ADAMBS is effective in reducing the training time for several tasks with different deep learning models.

## Acknowledgements

This material is based upon work supported by the National Science Foundation under Grant No. 1629397 and the Michigan Institute for Data Science (MIDAS) PODS. The authors would like to thank Junghwan Kim and Morgan Lovay for their detailed feedback on the manuscript, and anynomous reviewers for their insightful comments.

## Broader Impact

As machine learning techniques are being used in more and more real-life products, deep learning is the most notable driving force behind it. Deep learning models have achieved state-of-the-art performance in scenarios such as image recognition, natural language processing, and so on. Our society has benefited greatly from the success of deep learning models. However, this success normally relies on large amount of data available to train the models using optimization methods such as Adam. In this paper, we propose a generalization of Adam that can be more efficient to train models on large amount of data, especially when the datasets are imbalanced. We believe our method could become a widely adopted optimization method for training deep learning models, thus bringing broad impact to many real-life products that rely on these models.

## Footnotes

[1] All omitted proofs can be found in the appendix.

[2]Please see the appendix for curves of training error rate vs. wall clock time.

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
