[Supplementary Material]

# Appendix to "Adam with Bandit Sampling for Deep Learning"

**Rui Liu, Tianyi Wu, Barzan Mozafari**
Computer Science and Engineering
University of Michigan, Ann Arbor
{ruixliu, tianyiwu, mozafari}@umich.edu

## A  Proof of Theorem 1

According to Theorem 4.1 in [1], the convergence rate of Adam is

$$
\sum_{t=1}^{T}[f_t(\theta_t) - f_t(\theta^*)]
$$

$$
\leq \frac{D^2}{2\alpha(1-\beta_1)} \sum_{i=1}^{d} \sqrt{T\hat{v}_{T,i}} +
$$

$$
\frac{\alpha(1+\beta_1)G_\infty}{(1-\beta_1)\sqrt{1-\beta_2}(1-\gamma)^2} \sum_{i=1}^{d} \|g_{1:T,i}\|_2 +
$$

$$
\sum_{i=1}^{d} \frac{D_\infty^2 G_\infty \sqrt{1-\beta_2}}{2\alpha(1-\beta_1)(1-\lambda)^2}
\tag{1}
$$

First, we show $\sum_{i=1}^{d}\sqrt{\hat{v}_{T,i}} \leq dG_\infty$

*Proof.*

$$
\sum_{i=1}^{d}\sqrt{\hat{v}_{T,i}}
$$

$$
= \sum_{i=1}^{d}\sqrt{\sum_{j=1}^{T}\frac{1-\beta_2}{1-\beta_2^T}\beta_2^{T-j}g_{j,i}^2}
$$

$$
\leq \sum_{i=1}^{d}\sqrt{\sum_{j=1}^{T}\frac{1-\beta_2}{1-\beta_2^T}\beta_2^{T-j}G_\infty^2}
\tag{2}
$$

$$
= G_\infty\sqrt{\frac{1-\beta_2}{1-\beta_2^T}}\sum_{i=1}^{d}\sqrt{\sum_{j=1}^{T}\beta_2^{T-j}}
$$

$$
= G_\infty\sqrt{\frac{1-\beta_2}{1-\beta_2^T}}\sum_{i=1}^{d}\sqrt{\frac{1-\beta_2^T}{1-\beta_2}}
$$

$$
= dG_\infty
$$

□

Therefore, the above bound can be rewritten as

$$\sum_{t=1}^{T}[f_t(\theta_t) - f_t(\theta^*)]$$

$$\leq \rho_1 d\sqrt{T} + \sqrt{d}\rho_2\sqrt{\sum_{t=1}^{T}\|\hat{G}_t\|^2} + \rho_3 \tag{3}$$

where

$$\rho_1 = \frac{D^2 G_\infty}{2\alpha(1 - \beta_1)}$$

$$\rho_2 = \frac{\alpha(1 + \beta_1)G_\infty}{(1 - \beta_1)\sqrt{1 - \beta_2}(1 - \gamma)^2} \tag{4}$$

$$\rho_3 = \sum_{i=1}^{d} \frac{D_\infty^2 G_\infty \sqrt{1 - \beta_2}}{2\alpha(1 - \beta_1)(1 - \gamma)^2}$$

*Proof.*

$$\sum_{t=1}^{T}[f_t(\theta_t) - f_t(\theta^*)]$$

$$\leq \rho_1 d\sqrt{T} + \rho_2 \sum_{i=1}^{d}\|g_{1:T,i}\|_2 + \rho_3$$

$$= \rho_1 d\sqrt{T} + d\rho_2 \sum_{i=1}^{d}\frac{1}{d}\sqrt{\sum_{t=1}^{T}(\hat{G}_{t,i})^2} + \rho_3 \tag{5}$$

$$\text{(due to } \sqrt{\cdot} \text{ is concave)}$$

$$\leq \rho_1 d\sqrt{T} + d\rho_2\sqrt{\sum_{i=1}^{d}\frac{1}{d}\sum_{t=1}^{T}(\hat{G}_{t,i})^2} + \rho_3$$

$$= \rho_1 d\sqrt{T} + \sqrt{d}\rho_2\sqrt{\sum_{t=1}^{T}\|\hat{G}_t\|^2} + \rho_3$$

$\square$

we could get the following theorem for the convergence of Adam with sampling with replacement with batch size $K$.

$$\sum_{t=1}^{T}[f_t(\theta_t) - f_t(\theta^*)]$$

$$= \rho_1 d\sqrt{T} + \sqrt{d}\rho_2\sqrt{\frac{1}{n^2 K}\sum_{t=1}^{T}\mathbb{E}\left[\sum_{j=1}^{n}\frac{\|g_t\|^2}{p_j^t}\right]} + \rho_3 \tag{6}$$

*Proof.* Since $\hat{G}_t = \frac{1}{K}\sum_{k=1}^{K}\hat{g}_{I_k^t} = \frac{1}{K}\sum_{k=1}^{K}\frac{g_{I_k^t}}{np_{I_k^t}}$, we have

$$
\begin{aligned}
\mathbb{E}[\|\hat{G}_t\|^2] \leq & \frac{1}{n^2K^2}\sum_{k=1}^{K}\mathbb{E}\left[\frac{\|g_{I_k^t}\|^2}{p_{I_k^t}^2}\right] \\
= & \frac{1}{n^2K^2}\sum_{k=1}^{K}\mathbb{E}\left[\sum_{j=1}^{n}\frac{\|g_j^t\|^2}{(p_j^t)^2}p_j^t\right] \\
= & \frac{1}{n^2K^2}\sum_{k=1}^{K}\mathbb{E}\left[\sum_{j=1}^{n}\frac{\|g_j^t\|^2}{p_j^t}\right] \\
= & \frac{1}{n^2K}\mathbb{E}\left[\sum_{j=1}^{n}\frac{\|g_j^t\|^2}{p_j^t}\right]
\end{aligned}
\tag{7}
$$

From previous step, we know

$$
\begin{aligned}
& \sum_{t=1}^{T}[f_t(\theta_t) - f_t(\theta^*)] \\
\leq & \rho_1 d\sqrt{T} + \sqrt{d}\rho_2\sqrt{\sum_{t=1}^{T}\|\hat{G}_t\|^2} + \rho_3 \\
\leq & \rho_1 d\sqrt{T} + \sqrt{d}\rho_2\sqrt{\sum_{t=1}^{T}\frac{1}{n^2K}\mathbb{E}\left[\sum_{j=1}^{n}\frac{\|g_j^t\|^2}{p_j^t}\right]} + \rho_3 \\
= & \rho_1 d\sqrt{T} + \sqrt{d}\rho_2\sqrt{\frac{1}{n^2K}\sum_{t=1}^{T}\mathbb{E}\left[\sum_{j=1}^{n}\frac{\|g_j^t\|^2}{p_j^t}\right]} + \rho_3
\end{aligned}
\tag{8}
$$

$\square$

## B Proof of Theorem 2

Theorem 2 follows from Theorem 1 and Lemma 1. Therefore, we focus on the proof of Lemma 1 here. We prove Lemma 1 using the framework of online learning with bandit feedback.

Online optimization is interested in choosing $p^t$ to solve the following problem

$$
\min_{p^t \in \mathcal{P}, 1 \leq t \leq T}\sum_{t=1}^{T}L_t(p^t)
\tag{9}
$$

where $L_t(p^t)$ is the loss that incurs at each iteration. Equivalently, the goal is the same as minimizing the pseudo-regret:

$$
\bar{R}_T = \mathbb{E}\sum_{t=1}^{T}L_t(p^t) - \min_{p \in \mathcal{P}}\mathbb{E}\sum_{t=1}^{T}L_t(p)
\tag{10}
$$

The following Algorithm 1 similar to EXP3 could be used to solve the above problem.

To be clear, the Bregman divergence $B_\phi(x, y) = \phi(x) - \phi(y) - \langle\nabla\phi(y), x - y\rangle$. Note that, the updating step of Algorithm 1 is equivalent to

$$
\begin{aligned}
w^{t+1} &= \nabla\phi^*(\nabla\phi(p^t) - \alpha_p\hat{h}_t) \\
p^{t+1} &= \underset{y \in \mathcal{P}}{\arg\min}\, B_\phi(y, w^{t+1})
\end{aligned}
\tag{11}
$$

We have the following convergence result for Algorithm 1.

**Algorithm 1** Online optimization
---
1: Input: stepsize $\alpha_p$
2: Initialize $x^1$ and $p^1$.
3: **for** $t = 1, \cdots, T$ **do**
4:     Play a perturbation $\tilde{p}^t$ of $p^t$ and observe $\nabla L_t(\tilde{p}^t)$
5:     Compute an unbiased gradient estimate $\hat{h}_t$ of $\nabla L_t(p^t)$, i.e. as long as $\mathbb{E}[\hat{h}_t] = \nabla L_t(p^t)$
6:     Update $p^t$: $p^{t+1} \leftarrow argmin_{p \in \mathcal{P}} \left\{ \langle \hat{h}_t, p \rangle + \frac{1}{\alpha_p} B_\phi(p, p^t) \right\}$.
7: **end for**
---

**Proposition B.1.** *The Algorithm 1 has the following convergence result*

$$\bar{R}_T = \mathbb{E} \sum_{t=1}^{T} L_t(\tilde{p}^t) - \min_{p \in \mathcal{P}} \mathbb{E} \sum_{t=1}^{T} L_t(p)$$

$$\leq \frac{B_\phi(p, p^1)}{\alpha_p} + \frac{1}{\alpha_p} \sum_{t=1}^{T} \mathbb{E} B_{\phi^*}(\nabla \phi(p^t) - \alpha_p \hat{h}_t, \nabla \phi(p^t)) \qquad (12)$$

$$+ \sum_{t=1}^{T} \mathbb{E} \left[ \|p_t - \tilde{p}_t\| \|\hat{h}_t\|_* \right]$$

*If $L_t(p^t)$ is linear (i.e. $L_t(p^t) = \langle l_t, p^t \rangle$), then we have*

$$\bar{R}_T = \mathbb{E} \sum_{t=1}^{T} L_t(\tilde{p}^t) - \min_{p \in \mathcal{P}} \mathbb{E} \sum_{t=1}^{T} L_t(p)$$

$$\leq \frac{B_\phi(p, p^1)}{\alpha_p} + \frac{1}{\alpha_p} \sum_{t=1}^{T} \mathbb{E} B_{\phi^*}(\nabla \phi(p^t) - \alpha_p \hat{h}_t, \nabla \phi(p^t)) \qquad (13)$$

$$+ \sum_{t=1}^{T} \mathbb{E} \left[ \|p_t - \mathbb{E}[\tilde{p}_t | p_t]\| \|\hat{h}_t\|_* \right]$$

For example, let $\phi(p) = \sum_{j=1}^{n} p_j \log p_j$. Then its convex conjugate is $\phi^*(u) = \sum_{j=1}^{n} \exp(u_j - 1)$. In this case, due to Equation (11), the updating step of Algorithm 1 becomes

$$w_j^{t+1} = p_j^t \exp(-\alpha_p \hat{h}_{t,j}), \forall 1 \leq j \leq n \qquad (14)$$

The convergence result can also be simplified because

$$B_{\phi^*}(\nabla \phi(p^t) - \alpha_p \hat{h}_t, \nabla \phi(p^t))$$

$$= \sum_{j=1}^{n} p_j^t \left( \exp(-\alpha_p \hat{h}_{t,j}) + \alpha_p \hat{h}_{t,j} - 1 \right)$$

$$\text{(assume } \hat{h}_{t,j} \geq 0 \text{ and due to } e^z - z - 1 \leq z^2/2 \text{ for } z \leq 0 \text{)} \qquad (15)$$

$$\leq \frac{\alpha_p^2}{2} \sum_{j=1}^{n} p_j^t \hat{h}_{t,j}^2$$

**Linear Case:** Let's consider a special case where $L_t(p^t)$ is linear, i.e. $L_t(p^t) = \langle l_t, p^t \rangle$. Assume $p^t$ is a probability distribution, i.e. $\mathcal{P} = \{p : \sum_{i=1}^{n} p_i = 1, 0 \leq p_i \leq 1, \forall i\}$. In this case, $\nabla L_t(p^t) = l_t$. At iteration $t$, assume that we can't get the whole vector of $l_t$. Instead, we can get only one coordinate $l_{t,J_t}$, where $J_t$ is sampled according to the distribution $p^t$. This is equivalent to $\tilde{p}_j^t = \mathbb{1}(j = J_t)$. Obviously,

$$\mathbb{E}[\tilde{p}_j^t] = \mathbb{E}[\mathbb{1}(j = J_t)] = \sum_{i=1}^{n} p_i^t \mathbb{1}(j = i) = p_j^t \qquad (16)$$

Furthermore, to get the unbiased estimate of gradient, we set $\hat{h}_{t,j} = \frac{l_{t,j}\mathbb{1}(j=J_t)}{p_j^t}$. To verify $\hat{h}_{t,j}$ is indeed unbiased w.r.t. $l_{t,j}$,

$$\mathbb{E}[\hat{h}_{t,j}] = \mathbb{E}[\frac{l_{t,j}\mathbb{1}(j=J_t)}{p_j^t}] = \frac{l_{t,j}\mathbb{E}[\mathbb{1}(j=J_t)]}{p_j^t} = \frac{l_{t,j}}{p_j^t}p_j^t = l_{t,j} \tag{17}$$

For sampling multiple actions $I^t$, we have the following

$$\hat{h}_{t,j} = \frac{l_{t,j}\sum_{k=1}^{K}\mathbb{1}(j=I_k^t)}{Kp_j^t} \tag{18}$$

And its convergence result is

$$\bar{R}_T = \mathbb{E}\sum_{t=1}^{T}L_t(\tilde{p}^t) - \min_{p\in\mathcal{P}}\mathbb{E}\sum_{t=1}^{T}L_t(p)$$

$$\leq \frac{B_\phi(p,p^1)}{\alpha_p} + \frac{1}{\alpha_p}\sum_{t=1}^{T}\mathbb{E}B_{\phi^*}(\nabla\phi(p^t) - \alpha_p\hat{h}_t, \nabla\phi(p^t))$$

$$+ \sum_{t=1}^{T}\mathbb{E}\left[\|p_t - \mathbb{E}[\tilde{p}_t|p_t]\|\|\hat{h}_t\|_*\right]$$

$$= \frac{B_\phi(p,p^1)}{\alpha_p} + \frac{1}{\alpha_p}\sum_{t=1}^{T}\mathbb{E}B_{\phi^*}(\nabla\phi(p^t) - \alpha_p\hat{h}_t, \nabla\phi(p^t))$$

$$\leq \frac{B_\phi(p,p^1)}{\alpha_p} + \frac{\alpha_p}{2}\sum_{t=1}^{T}\mathbb{E}\sum_{j=1}^{n}p_j^t\hat{h}_{t,j}^2 \tag{19}$$

$$= \frac{B_\phi(p,p^1)}{\alpha_p} + \frac{\alpha_p}{2}\sum_{t=1}^{T}\mathbb{E}\sum_{j=1}^{n}p_j^t\frac{l_{t,j}^2(\sum_{k=1}^{K}\mathbb{1}(j=I_k^t))^2}{(Kp_j^t)^2}$$

$$\leq \frac{B_\phi(p,p^1)}{\alpha_p} + \frac{\alpha_p}{2}\sum_{t=1}^{T}\mathbb{E}\sum_{j=1}^{n}p_j^t\frac{l_{t,j}^2\sum_{k=1}^{K}(\mathbb{1}(j=I_k^t))^2}{K(p_j^t)^2}$$

$$= \frac{B_\phi(p,p^1)}{\alpha_p} + \frac{\alpha_p}{2}\sum_{t=1}^{T}\mathbb{E}\frac{l_{t,J_t}^2}{p_{J_t}^t}$$

$$= \frac{B_\phi(p,p^1)}{\alpha_p} + \frac{\alpha_p}{2}\sum_{t=1}^{T}\sum_{j=1}^{n}l_{t,j}^2$$

We want to apply online optimization/bandit to learn $p_j^t$, in which case the loss at $t$-th iteration is $l_t(p^t) = \sum_{j=1}^{n}\frac{\|g_j(x^t)\|_*^2}{p_j^t}$. Because the loss $l_t(p^t)$ is a nonlinear function, we could use nonlinear bandit as a general approach to solve this. However, another simpler approach is to convert it to linear problem, where linear bandit could be used. Specifically, we have

$$\sum_{t=1}^{T}\mathbb{E}\left[\sum_{j=1}^{n}\|g_j(x^t)\|_*^2\left(\frac{1}{p_j^t} - \frac{1}{p_j}\right)\right]$$

$$\leq \sum_{t=1}^{T}\mathbb{E}\left[\left\langle -\{\|g_j(x^t)\|_{j,*}^2/(p_j^t)^2\}_{j=1}^{n}, p^t - p\right\rangle\right] \tag{20}$$

$$= \sum_{t=1}^{T}\mathbb{E}\left[\left\langle -\left\{\|g_j(x^t)\|_{j,*}^2/(p_j^t)^2 + \frac{L^2}{p_{min}^2}\right\}_{j=1}^{n}, p^t - p\right\rangle\right]$$

This is equivalent to online linear optimization setting where $l_{t,j}(x) = -\frac{\|g_j(x)\|^2_{j,*}}{(p^t_j)^2} + \frac{L^2}{p^2_{min}}$. And it's easy to see that $0 \le l_{t,j} \le \frac{L^2}{p^2_{min}}$. Then the convergence result is

$$
\begin{aligned}
&\mathbb{E}\sum_{t=1}^{T} L_t(\tilde{p}^t) - \min_{p \in \mathcal{P}} \mathbb{E}\sum_{t=1}^{T} L_t(p) \\
&= \frac{B_\phi(p, p^1)}{\alpha_p} + \frac{\alpha_p}{2} \sum_{t=1}^{T}\sum_{j=1}^{n} l^2_{t,j} \\
&\le \frac{B_\phi(p, p^1)}{\alpha_p} + \frac{\alpha_p}{2} \sum_{t=1}^{T}\sum_{j=1}^{n} \frac{L^4}{p^4_{min}} \\
&= \frac{B_\phi(p, p^1)}{\alpha_p} + \frac{\alpha_p}{2} \frac{nTL^4}{p^4_{min}} \\
&\le \frac{R^2}{\alpha_p} + \frac{\alpha_p}{2} \frac{nTL^4}{p^4_{min}} \\
&\quad (\text{set } \alpha_p = \sqrt{\frac{2R^2 p^4_{min}}{nTL^4}}) \\
&= \frac{RL^2}{p^2_{min}} \sqrt{2nT}
\end{aligned}
\tag{21}
$$

## C Proofs about Comparison with Uniform Sampling

### C.1 Proof of Lemma 2

*Proof.*

$$
\begin{aligned}
&\mathbb{E}\sum_{j=1}^{n} \frac{\|g_j\|^2}{p_j} \\
&= \mathbb{E}n \sum_{j=1}^{n} \|g_j\|^2 \\
&= n \sum_{j=1}^{n} \mathbb{E}\|g_j\|^2 \\
&\le n \sum_{j=1}^{n} \mathbb{E}\sum_{i=1}^{d} z^2_{j,i} \\
&= n \sum_{j=1}^{n}\sum_{i=1}^{d} \beta_3 i^{-\gamma} j^{-\gamma} \\
&= \beta_3 n \sum_{j=1}^{n} j^{-\gamma} \sum_{i=1}^{d} i^{-\gamma} \\
&\le \beta_3 n \log n \log d
\end{aligned}
\tag{22}
$$

□

## C.2 Proof of Theorem 3

*Proof.* By plugging Lemma 2 into Theorem 2, we have

$$
\sum_{t=1}^{T}[f_t(\theta_t) - f_t(\theta^*)]
$$
$$
\leq \rho_1 d\sqrt{T} + \frac{\rho_2\sqrt{\beta_3}}{\sqrt{K}}\sqrt{d\log d}\frac{\sqrt{n\log n}}{n}\sqrt{T} + \rho_3
$$
(23)

$\square$

## C.3 Proof of Lemma 3

*Proof.*

$$
\min_{p\geq p_{min}}\sum_{j=1}^{n}\mathbb{E}\frac{\|g_j\|^2}{p_j}
$$
$$
= \min_{p\geq p_{min}}\sum_{j=1}^{n}\frac{1}{p_j}\mathbb{E}\sum_{i=1}^{d}z_{j,i}^2
$$
$$
= \min_{p\geq p_{min}}\sum_{j=1}^{n}\frac{1}{p_j}\sum_{i=1}^{d}\beta_3 i^{-\gamma}j^{-\gamma}
$$
$$
= \min_{p\geq p_{min}}\sum_{j=1}^{n}\beta_3\frac{j^{-\gamma}}{p_j}\sum_{i=1}^{d}i^{-\gamma}
$$
$$
\leq \min_{p\geq p_{min}}\sum_{j=1}^{n}\beta_3\frac{j^{-\gamma}}{p_j}\log d
$$
$$
= \beta_3\log d\min_{p\geq p_{min}}\sum_{j=1}^{n}\frac{j^{-\gamma}}{p_j}
$$
(24)

From Proposition 5 in [2], we know

$$
\min_{p\geq p_{min}}\sum_{j=1}^{n}\frac{j^{-\gamma}}{p_j} = O(\log^2 n)
$$
(25)

Thus, we have

$$
\min_{p\geq p_{min}}\sum_{j=1}^{n}\mathbb{E}\frac{\|g_j\|^2}{p_j}
$$
$$
= \beta_3\log d\min_{p\geq p_{min}}\sum_{j=1}^{n}\frac{j^{-\gamma}}{p_j}
$$
$$
= O(\log d\log^2 n)
$$
(26)

$\square$

## C.4 Proof of Theorem 4

*Proof.* It follows simply by plugging Lemma 3 into Theorem 2. $\square$

# D   AMSGrad with Bandit Sampling

We can also use bandit sampling to endow AMSGrad [3] with the ability to adapt to different training examples. Analogous to Theorem 1 in the main text, we have the following theorem regarding AMSGrad with Bandit Sampling.

**Theorem D.1.** *Assume the gradient $\hat{G}_t$ is bounded, $\|\hat{G}_t\|_\infty \le G_\infty$, and $\alpha_t = \alpha/\sqrt{t}$, $\beta_1 = \beta_{11}$, and $\gamma = \beta_1/\sqrt{\beta_2} < 1$, $\beta_{1t} = \beta_1 \lambda^{t-1}$. Then, AMSGrad with Bandit Sampling achieves the following convergence rate*

$$R(T) \le \rho_1' d\sqrt{T} + \rho_2' \sqrt{d(1 + \log T)} \sqrt{\sum_{t=1}^T \|\hat{G}_t\|^2} + \rho_3'. \tag{27}$$

*where*

$$
\begin{aligned}
\rho_1' =& \frac{D_\infty^2 G_\infty}{\alpha(1 - \beta_1)} \\
\rho_2' =& \frac{\alpha}{(1 - \beta_1)^2(1 - \gamma)\sqrt{1 - \beta_2}} \\
\rho_3' =& \frac{\beta_1 D_\infty^2 G_\infty}{2(1 - \beta_1)(1 - \lambda)^2}
\end{aligned}
\tag{28}
$$

*Proof.* According to Corrollary 1 from [3], we have

$$R(T) \le \frac{\rho_1'\sqrt{T}}{G_\infty} \sum_{i=1}^d \hat{v}_{T,i}^{1/2} + \rho_2' \sqrt{(1 + \log T)} \sum_{i=1}^d \|g_{1:T,i}\|_2 + \rho_3'. \tag{29}$$

From section "Proof of Theorem 1", we know that

$$\sum_{i=1}^d \sqrt{\hat{v}_{T,i}} \le dG_\infty \tag{30}$$

Therefore,

$$
\begin{aligned}
R(T) \le& \rho_1' d\sqrt{T} + \rho_2' \sqrt{1 + \log T} \sum_{i=1}^d \|g_{1:T,i}\|_2 + \rho_3' \\
=& \rho_1' d\sqrt{T} + d\rho_2' \sqrt{1 + \log T} \sum_{i=1}^d \frac{1}{d} \sqrt{\sum_{t=1}^T (\hat{G}_{t,i})^2} + \rho_3' \\
& \text{(due to } \sqrt{\cdot} \text{ is concave)} \\
\le& \rho_1' d\sqrt{T} + d\rho_2' \sqrt{1 + \log T} \sqrt{\sum_{i=1}^d \frac{1}{d} \sum_{t=1}^T (\hat{G}_{t,i})^2} + \rho_3' \\
=& \rho_1' d\sqrt{T} + \sqrt{d(1 + \log T)} \rho_2' \sqrt{\sum_{t=1}^T \|\hat{G}_t\|^2} + \rho_3'
\end{aligned}
\tag{31}
$$

$\square$

Simiarly, we could derive theorems about AMSGrad with Bandit Sampling that are analogous to Theorem 2, 3, 4 for ADAMBS. We omit the details here.

## E    Additional Experiments

### E.1    Comparison with Basic SGD Variants

In the main paper, we compared our method with Adam and Adam with importance sampling. Here, we compare with some basic SGD variants (e.g., SGD, Momentum, Adagrad and RMSprop). As shown in Figure 1, these basic SGD variants have worse performance than Adam-based methods.

(a) CNN on CIFAR10  (b) RCNN on IMDB

Figure 1: Experiments comparing with SGD variants.

## E.2 Comparison on Error Rate

In the main paper, we have shown the plots of loss value vs. wall clock time. Some might also be interested in the convergence curves of loss values. Here, we include some plots of error rate vs. wall clock time, as shown in Figure 2 and Figure 3. They demonstrated the faster convergence of our method in terms of the error rate.

(a) CNN on CIFAR10  (b) CNN on CIFAR100

(c) CNN on Fashion MNIST

Figure 2: Error Plot of CNN models

(a) RNN on Fashion MNIST  (b) RCNN on IMDB

Figure 3: Error Plot of RNN and RCNN models