[Reviews · NeurIPS 2020]

Review 1

Summary and Contributions: This work proposes an extension of ADAM named ADAMBS which utilizes the bandit sampling for training examples. An obtained convergence rate of the proposed method is significantly better than that of ADAM in terms of the number of training examples, and the empirical speed-up also verified for learning CNNs, RNNs, and RCNNs. ---- After reading the response. I appreciate your hard work. I would like to raise the score.

Strengths: An improvement of the dependence on the number of training examples in the convergence rate is remarkable as well as the better empirical performance.

Weaknesses: It would be nice to compare the proposed method with other competitors (SGD, momentum, Adagrad, RMSprop, ...).

Correctness: The claims seem correct.

Clarity: The paper is well written and easy to read.

Relation to Prior Work: The relationship with related studies is well discussed.

Reproducibility: No

Additional Feedback: It would be nice if the authors could conduct additional experiments to veryfy the effectiveness of the proposed methods over existing methods (e.g., SGD, momentum, Adagrad, RMSprop)


Review 2

Summary and Contributions: The authors proposed a generalization of the ADAM optimizer for deep learning by sampling training examples according to their importance to the model's convergence. The sampling is performed using a multi-armed bandit algorithm. The authors showed that the new optimizer, ADAMBS, improves the convergence rate both theoretically and practically on different benchmarks.

Strengths: 1. The authors demonstrated that the proposed optimizer Adambs significantly improves both the convergence speed and the optimal value achieved in multiple benchmarks. Given that the computational complexity for machine learning tasks are becoming the acute bottleneck as we are training ever more complex model with huge amount of data, this is a significant result; 2. In addition to practical improvements, the authors also showed that in a simple setting, adding bandit sampling to Adam can improve the regret w.r.t. n, the number of training samples, by a factor of O(\sqrt(n/log n)). This is a significant speedup if the assumptions are satisfied since n can be a very big number in practice. 3. The presentation is clear with good reference to prior work, making it easy to identify the novelty and significance of the paper.

Weaknesses: 1. The comparison between ADAM and ADAMBS where the authors showed superior convergence is limited to a specific setting and assumed the feature vector follows a doubly heavy tailed distribution. It is unclear how important the doubly heavy tailed distribution assumption is to the result, and not very clear whether this assumption holds in practice. 2. While there is a significant improvement in terms of n, there is no difference in terms of d, which is the size of the feature vector. If we are training models where there are more parameters than training example, then the proposed method would not make significant difference compared to Adam.

Correctness: Yes.

Clarity: Yes.

Relation to Prior Work: Yes.

Reproducibility: Yes

Additional Feedback:


Review 3

Summary and Contributions: This work proposes ADAMBS, an attempt towards improving convergence of ADAM. The proposed procedure is based upon sampling minibatches during training that can aid in convergence. The sampling distribution at each iteration is updated through a multi-armed bandit procedure. This work provides both theoretical analysis and empirical explorations to support their claim.

Strengths: The claims seem to be theoretically sound. The paper also provides empirical support for the claims. The work does have some novelty. The use of multi-arm bandits for learning sampling distributions at each iteration of ADAM seems useful (and seems novel in the context of ADAM). The mixture of ADAM and multi-arm bandits can motivate the broader NeurIPS community to apply/explore/expand on this approach for gradient based optimization problems (that have a very wide usage). Therefore, it does seem relevant to the broader NeurIPS community.

Weaknesses: The paper proposes an approach for sampling minibatches. It, however, does not explore the strength of this proposed approach in a more broader manner. Is there something specific to ADAM that makes the sampling approach more useful. Furthermore, setups such as curriculum learning can also perhaps provide adaptive minibatches during training. It does not seem that such avenues were explored as comparisons to the proposed setup.

Correctness: The claims seem to be correct. Standard/Known methodology seems to have been used during empirical explorations

Clarity: The paper does seem to be clearly written.

Relation to Prior Work: This work does discusses prior work in the area and tries to show how it is different from them.

Reproducibility: Yes

Additional Feedback: This work seems to propose an approach for sampling minibatches that can perhaps be applied to other procedures apart from ADAM. Therefore, apart from ADAM, was this approach (or suitable variants) explored (perhaps empirically) for other optimiztions procedures that involve minibatch? or is it that the proposed sampling approach is suitable only for ADAM-like procedures? What about curriculum learning? It can also be used to produce desired minibatches for better training. How does this approach compare to the state of the art in curriculum learning. In Algorithm 2 Line 2, what is L ? What is the influence of L and p_min in Algorithm 2? It seems that ADAMBS seems to converge to a lower loss value in most presented empirical explorations. Is that a typical trend or just an observation for the given data sets and architectures. If it is a common trend then was the influence of the sampling procedure on this phenomenon explored? perhaps minor: in line 99: f is dependent on the data. Therefore it is a function of more than just \theta. ##### after rebuttal ######## Thanks for the clarifications. ########################

[Author Response · NeurIPS 2020]

1 We sincerely thank our reviewers for their insightful comments. Below we address all their comments, numbered as
2 C1-C7 (corresponding answers are numbered as A1-A7).

3 *Reviewer 1*
4 C1: It would be nice if the authors could conduct additional experiments to verify the effectiveness of the proposed
5 methods over existing methods (e.g., SGD, momentum, Adagrad, RMSprop)
6 A1: We thank the reviewer for mentioning these methods. We have conducted experiments with these competitors. We
7 didn't include them because they have worse performance than Adam in our experiments, which have been shown in
8 the original Adam paper [16]. Due to the space limit, we include the results on two datasets (CIFAR10 and IMDB) in
9 Figure 1 as below, and will happily include them in our camera-ready if accepted.

(a) CNN on CIFAR10    (b) RCNN on IMDB

Figure 1: Additional experiments

*Reviewer 2*

C2: It's unclear how important the doubly heavy tailed distribution assumption is to the result, and whether this assumption holds in practice.

A2: We thank the reviewer for the comment. The key intuition behind the design of our algorithm is to be adaptive to the difference among the examples. Our method is therefore particularly useful when the dataset is imbalanced. The doubly heavy tailed distribution is

22 just one example of imbalanced data that we used for convergence analysis, as its simplicity facilitates the convergence
23 analysis. Doubly-heavy tail distribution is not a single distribution, rather a family of distributions parameterized by
24 the value of $\gamma$. As for its importance, doubly heavy tailed distribution is primarily responsible for getting the $\log(n)$
25 speedup. For other non-uniform distributions, the convergence will still be faster than Adam, but the speedup may be
26 lower or higher than $\log(n)$ depending on the particular parameters of the distribution.

27 C3: If we are training models where there are more parameters than training examples, then the proposed method would
28 not make significant difference compared to Adam.
29 A3: We thank the reviewer for bringing this up. We fully agree that our method shines when the number of training
30 examples is massive. This is particularly aligned with the current trend in deep learning to use more and more data for
31 training. As for the model size issue, we'd like to note that significant efforts in the community have been dedicated
32 to reducing the model size because deep learning models are often over-parameterized. Our method is orthogonal,
33 and thus we think combined with these model-size reduction methods, it will continue to retain its benefits due to its
34 alignment with existing trends in the industry and the research community.

35 *Reviewer 3*
36 C4: is it that the proposed sampling approach is suitable only for ADAM-like procedures?
37 A4: The reviewer is correct. The idea behind our approach can be extended to other optimization procedures. We chose
38 Adam due to its superior empirical performance and its well-understood theoretical properties. Exploring this extension
39 to other minibatch-based optimizations will make an interesting direction of future research.

40 C5: What about curriculum learning? How does this approach compare to the state of the art in curriculum learning.
41 A5: We thank the reviewer for mentioning curriculum learning. Curriculum learning is indeed an important optimization
42 strategy, which leverages a pre-trained teacher model to train the target model. The teacher model is critical in
43 determining mini-batches. We will happily include a discussion and empirical experiments comparing with curriculum
44 learning in our camera-ready if accepted.

45 C6: In Algorithm 2 Line 2, what is $L$? What is the influence of $L$ and $p_{min}$ in Algorithm 2?
46 A6: We apologize for not having made this clear. The $L$ in Algorithm 2 is the upper bound on the gradient norm
47 (formally defined in Lemma 1), which is commonly assumed in related literature. This assumption holds in most cases,
48 especially when gradient clipping trick is applied. The influence of $L$ and $p_{min}$ is related to the convergence analysis of
49 the bandit method. Simply speaking, they make sure the loss $l_{t,j}$ is always nonnegative, ensuring the correctness of
50 distribution update.

51 C7: It seems that ADAMBS ... a lower loss value ... Is that a typical trend ...? If it is a common trend then was the
52 influence of the sampling procedure on this phenomenon explored?
53 A7: We think this is a common trend, due to the sampling procedure of our method. The reason is that our method
54 tends to sample examples with large gradient norms at each iteration, which effectively reduces gradient variance as
55 discussed in Section 4.2. Some experiments have shown a similar trend for variance reduction techniques.

[Meta-Review · NeurIPS 2020]

Authors propose a method for adaptive selection of data points for SGD. Specifically, authors use the ADAM method and extend it to adaptive sampling setting using multi-armed bandit. Proposed method is further analyzed and improvement in the convergence speed is quantified. Extensive empirical results also support the proposed method. All reviewers unanimously recommend accept. Clear accept.